# Estimating the Impact of Ecological Migrants on the South-to-North Water Diversion in China

**DOI:** 10.3390/ijerph182312295

**Published:** 2021-11-23

**Authors:** Mengdi Li, Yaoping Cui, Yaochen Qin, Zhifang Shi, Nan Li, Xiaoyan Liu, Yadi Run, Oliva Gabriel Chubwa

**Affiliations:** 1Key Laboratory of Geospatial Technology for the Middle and Lower Yellow River Regions, Henan University, Ministry of Education, Kaifeng 475004, China; lmd@henu.edu.cn (M.L.); shizhifang@henu.edu.cn (Z.S.); linan0716@henu.edu.cn (N.L.); lxy@henu.edu.cn (X.L.); run@henu.edu.cn (Y.R.); oliva2020@vip.henu.edu.cn (O.G.C.); 2College of Geography and Environmental Science, Henan University, Kaifeng 475004, China; 3Henan Science and Technology Innovation Center of Natural Resources, Zhengzhou 450008, China; 4Dabieshan National Observation and Research Field Station of Forest Ecosystem, Henan University, Xinyang 464000, China

**Keywords:** South-to-North Water Diversion, ecological migrants, water ecosystem expansion, socioeconomic effect, GEE

## Abstract

The South-to-North Water Diversion (SNWD) provides significant benefits in facilitating water security and improving ecology in northern China. However, few studies have estimated the water value of the SNWD and the corresponding subsequent subsidies of the ecological migrants in Xichuan County displaced by the project. Based on the Google Earth Engine (GEE), this study analyzed the water ecosystem changes in Xichuan County in 2000–2020 and valued the water transfer of the SNWD. We calculated the water cost, the water value of the trunk line project, and the four provinces (Hebei, Henan, Beijing, and Tianjin) of CNY 4.04, 39.64, and 120.93 billion, respectively, and the proportion of the three was 1:10:30 during 2014–2020. The water ecosystem area showed a rapid increase when the SNWD became operational since the end of 2014. The subsequent annual subsidy gap of ecological migrants was CNY 0.84 billion, which only accounted for 4.31% of the gross profit of SNWD. Our results imply that relevant water sectors have sufficient profits to support corresponding subsequent subsidies for ecological migrants. Ecological migrants are a major challenge for water transfer projects. Overall, this study fills a gap of interactions between subsequent policies and ecological migrants and provides a typical case for managing the migration problem caused by sustainable water management worldwide.

## 1. Introduction

As populations and economies increase rapidly, water resource shortages have become a major global challenge [1,2]. With approximately 20% of the world’s population, and only approximately 5–7% of the global freshwater resources, China depends on groundwater withdrawal [3]. Groundwater offers approximately 60% and 70% of the total water supply in Beijing and Tianjin, respectively [3,4]. The reliance on groundwater has caused seawater intrusion in Tianjin [4]. Implementation of an extensive Chinese South-to-North Water Diversion (SNWD) began in 2002, intending to achieve the goal of clean water proposed by the United Nations and effectively alleviated the water shortage in northern China [5,6,7,8,9,10]. The SNWD includes west, middle, and east routes and will be able to transfer 44.8 billion m^3^ of freshwater each year to northern China if fully implemented by 2050 [11,12]. Previous studies showed that the contribution to groundwater recovery in Beijing by water replacement from SNWD was 40%, and the groundwater withdrawal decreased to 42% after four years from the SNWD became operational [13]. The decline of groundwater withdrawal in Beijing is attributed to the increasing replacement of a portion of groundwater withdrawal with the SNWD for domestic use and groundwater restrictions on agricultural water use by the government [13]. As a result, the groundwater level in Beijing rose from 25.7 m in 2014 to 22.7 m in 2019 [13]. By 2020 the SNWD had transported more than 25.8 billion m^3^ of water resources to northern China, accounting for more than 70% of all drinking water for the four provinces of Hebei, Henan, Beijing, and Tianjin through which it runs [8,14,15].

As water transfer from SNWD increased, the supplementation of ecological water became one of its important functions. The surface runoff of the four provinces has been supplied with an ecological water supplement, and the groundwater level has been restored with water from the SNWD [16,17,18]. The expansion of the water ecosystem brought some ecological benefits to the provinces along its line, significantly improving ecology and the local climate [12,19,20,21]. The SNWD plays an important role in promoting water security and improving the ecological environment, sustainable agricultural development, and social economy of northern cities [22,23,24]. It has also been a prominent example of Chinese capabilities and solutions for groundwater management and sustainable agricultural development worldwide [13,25].

The water use goal of the SNWD includes agricultural water and ecology water [15,26]. China has developed a two-part water price for the SNWD, which considers the cost of water transfer, project operation and maintenance, and user affordability [27]. The calculation of water price is based on dividing the product cost into the fixed cost and the variable cost and dividing the product price into the basic price (or capacity price) and the metering price. The water value was used for the operation and maintenance of the loan repayment for the SNWD. Most scholars have only analyzed the initial cost of water supply and the reasonableness of the water price set [28]. However, few studies have drawn on any systematic research into the total water value of the SNWD [29,30,31].

Ecological migrants refer to a situation where people lose their short-term or long-term survival benefits owing to deterioration of the environment, forcing them to change their place of residence [32,33]. The impact of development projects on ecological migrants could not be effectively solved within a short period because of the imperfect migrant policy, the low market development degree, and the undeveloped follow-up industry [34]. Researchers proposed that improving the education and training and developing industry could solve the above problems [34]. People are one of the most important components of an ecosystem, and the impact of relocation can be assessed subjectively [16,35]. It was reported that more than 300,000 people have been displaced because of the construction of the SNWD [36]. Previous studies on the impact of the SNWD have all focused on the environment and have ignored ecological migrants [37,38,39,40,41,42]. The resettlement of ecological migrants and examination of their living conditions are decisive factors in evaluating the impact of the SNWD [43,44]. The central line project of the SNWD originates from Danjiangkou reservoir with a long transfer distance and broad inundated land. In Xichuan County, the headwater source of the SNWD, more than 160,000 residents had to move away from their hometowns. Although ecological migrants were victims of national development and construction, insufficient attention has been given to corresponding subsequent subsidies paid to them.

This study aimed to estimate the water transfer value and the impact of ecological migrants of the SNWD. The specific objectives of this study were to: (1) estimate the total water cost and total water value using the water allocation data and water price data of the four provinces; (2) quantify the impact of ecological migrants combining the migrant statistics data, and the minimum standard of living for urban–rural residents data in the resettlement region; (3) provide scientific data for the relevant sectors on the water price and the subsequent subsidy policy regarding ecological migrants according to the estimated subsequent subsidy gap.

## 2. Materials and Methods

### 2.1. Study Area

The headwork county of the central line project of SNWD (32°55′ N–33°23′ N, 110°58′ E–111°53′ E) is Xichuan County, which is located in the southwest of Henan province on the western margin of Nanyang Basin, and adjacent to Hubei province. The largest artificial lake in Asia, Danjiangkou reservoir, is located in the south of Xichuan County. The southeast of Xichuan County is an area of alluvial plain and hills; the central area is hilly, and the northwest is an area of low mountains. The climate of Xichuan County belongs to the monsoon climate zone of transition from the northern subtropical zone to the warm temperate zone. Danjiang River runs through the whole study area from northwest to southeast, and its tributaries account for 93.5% of the total area of the county. The county includes 4 townships, 11 towns, and 2 sub-district offices and emerged from state poverty in 2020. Approximately 162,000 people have been relocated from Xichuan County to other counties and cities to ensure water quality in the SNWD headwork county and headwater source (Figure 1).

### 2.2. Data Source

In this study, Landsat TM surface reflectance data (https://earthexplorer.usgs.gov/, accessed on 22 November 2021). of Xichuan County from 2000 to 2020 were selected to extract the water ecosystem. We used farmland and urban–rural settlement ecosystem data of 2000, 2010, and 2020 from the Institute of Geographical Sciences and Resources of the Chinese Academy of Sciences (http://www.resdc.cn/, accessed on 22 November 2021) to analyze the relative changes in various ecosystem types during the study period. The outlet water price in each section of the central line project of SNWD is from the official website of the Construction and Administration Bureau of the South-to-North Water Diversion Middle Route Project (https://www.nsbd.cn/, accessed on 22 November 2021). Water price data (http://www.h2o-china.com/price/, accessed on 22 November 2021) of the four provinces (Hebei, Henan, Beijing, and Tianjin), since the SNWD became operational, were used to estimate the value of water transfer from 2014 to 2020. The official migrant statistics of Xichuan County were combined with the minimum standard of living data for urban–rural residents, issued by the governments in corresponding cities, to assess the impact of ecological migrants.

We used United States Geological Survey Landsat TM Collection 1 Tier 1 surface reflectance data from 2000 to 2020 (~1631 images) to identify the open surface water bodies in the study area (Figure 2). All the Landsat TM images in the study area, and extensive geospatial data sets from the National Aeronautics and Space Administration, were generated from the Google Earth Engine (GEE) (https://earthengine.google.org/, accessed on 22 November 2021). GEE is a cloud-based computing platform that provides high-performance computing capability. The Landsat Collection 1 Tier 1 images have been conducted with geometric and atmospheric corrections. We used the CF mask, a cloud masking method, to remove the cloud, cloud shadow, and snow pixels. The method performs well and is suitable for preparing Landsat data for change detection. All pixels remaining after pre-processing were collected as high-quality Landsat observations suitable for extracting surface water bodies.

### 2.3. Methods

#### 2.3.1. Water Ecosystem Extraction

The relationships between water and vegetation indices can be used to extract surface water bodies [45]. Using the water and vegetation indices, including modified normalized difference water index (*mNDWI*), enhanced vegetation index (*EVI*), and normalized difference vegetation index (*NDVI*), we extracted the water ecosystem for this study. Calculation of the water and vegetation indices was based on Equations (1)–(3):(1)$$mNDWI=\frac{\rho_{Green}-\rho_{SWIR1}}{\rho_{Green}+\rho_{SWIR1}}$$
(2)$$NDVI=\frac{\rho_{NIR}-\rho_{Red}}{\rho_{NIR}+\rho_{Red}}$$
(3)$$EVI=2.5\times \frac{\rho_{NIR}-\rho_{Red}}{1.0+\rho_{NIR}+6.0\rho_{Red}+7.5\rho_{Blue}}$$
where *ρ_Blue_*, *ρ_Green_*, *ρ_Red_*, *ρ_NIR_*, and *ρ_SWIR1_* are the surface reflectance values of blue (0.45–0.52 μm), green (0.52–0.60 μm), red (0.63–0.69 μm), near-infrared (*NIR*) (0.77–0.90 μm), and short-wave infrared-1 (*SWIR1*) (1.55–1.75 μm) bands in the Landsat TM sensors. We used *mNDWI* > *EVI* or *mNDWI* > *NDVI* to identify the pixels that showed a stronger water signal than vegetation signal. Vegetation pixels, mixed water, and vegetation pixels were removed by the criterion *EVI* < 0.1. Consequently, those pixels with the criteria [(*mNDWI* > *EVI* or *mNDWI* > *NDVI*) and (*EVI* < 0.1)] were classified as water body pixels, while other pixels were classified as non-water pixels [45,46]. We calculated the annual water frequency for each Landsat pixel in Xichuan County using Equation (4):(4)$$F\left(y\right)=\frac{1}{N_{y}}\overset{N_{y}}{\underset{i=1}{\sum }}w_{y,i}\times 100\%$$
where *F* denotes the water frequency of the pixel, *y* is the specific year, *N_y_* expresses the total Landsat TM observations of the pixel in that year, and *w_y,i_* represents whether the pixel is water. Water is indicated by 1, and non-water by 0. A permanent water body has water cover throughout the year [47]. We set 0.75 as the frequency threshold to extract permanent water bodies in our study, following earlier studies [45,46].

#### 2.3.2. Water Transfer

The central line project of SNWD delivers water to the cities of Nanyang, Pingdingshan, Xuchang, Zhengzhou, Jiaozuo, Xinxiang, Hebi, Anyang, Puyang, Luohe, Zhoukou, Dengzhou, Hua County, Handan, Xingtai, Shijiazhuang, Baoding, Langfang, Hengshui, Cangzhou, Xinji, Dingzhou, Beijing, and Tianjin. As previously mentioned, the two-part water price system adopted in the SNWD combines the basic price with the metering price. The basic water value and the metering water value reimburse the fixed cost and variable cost, respectively; that is, the basic water value repays the loan and reimburses the basic operational cost of the project, while the metering water value reimburses other reasonable costs. The metering water price is calculated according to the actual water use in each city. The basic water price is based on the distance from the SNWD. The further the water transfer distance, the higher the water price. The National Development and Reform Commission issued a notice on water supply price policy in the initial operation for the central line project of SNWD, and the outlet water price of the SNWD differed in various sections (Table 1).

We collected the water price in the various provinces from different industries and official government websites and found that the water prices of residents, non-residents, and particular industries are quite different. Therefore, this study used the first-tier water price (the lowest) data together with the water allocation data to calculate the water value of four provinces. The total water cost and the total water value of the trunk line project were calculated by the comprehensive water price of the headwater project and trunk line project combined with water allocation data, respectively. In addition, we defined the difference between the water value of the trunk line project and the water cost as the profits from China South-to-North Water Diversion Corporation Limited. The difference between the water value of the four provinces and the water value of the trunk line project is the profit from the water sectors in the four provinces. We calculated the total water cost, total water value of the trunk line project, and total water value of four provinces by the equations as follows:(5)$$\;\;\;TWC_{H}=\overset{m}{\underset{q=1}{\sum }}CWP_{H}WA_{q}$$
(6)$$TWV_{T}=\overset{m}{\underset{q=1}{\sum }}CWP_{T}WA_{q}$$
(7)$$TWV_{P}=\overset{m}{\underset{q=1}{\sum }}FWP_{P}WA_{q}$$
where the $TWC_{H}$ represents the total water cost; the $CWP_{H}$ is the comprehensive water price of the headwater project; the *WA* is the water allocation for each province; *q* represents the province in the line of SNWD; $TWV_{T}$ represents the total water value of the trunk line project; $CWP_{T}$ is the comprehensive water price of the trunk line project; $TWV_{P}$ is the total water value of the four provinces; $FWP_{P}$ is the average first-tier water price for residents of each province.

#### 2.3.3. Ecological Migrants

We used the water ecosystem extracted from the GEE and the farmland and urban–rural settlement ecosystem data of Xichuan County in 2000, 2010, and 2020 to analyze their relative changes. In addition, we also used official migrant data combined with the minimum standard of living for urban–rural residents to estimate the corresponding subsequent subsidy gap. All ecological migrants were included when calculating the minimum standard of living for urban–rural residents, and we calculated the corresponding gap in the subsequent annual subsidy for the Xichuan County ecological migrants according to Equation (8):

(8)$$S=\overset{k}{\underset{i=1}{\sum }}p_{i}n_{i}$$ where *S* represents the total gap in the subsequent subsidy for the ecological migrants, *p_i_* is the subsequent average subsidy to meet the minimum standard of living in urban–rural areas, *n_i_* represents the number of ecological migrants in each city, and *i* represents the city into which the ecological migrants move.

## 3. Results

### 3.1. Water Transfer of SNWD

According to the annual water allocation of SNWD from 2014 to 2020 (Figure 3), the annual water allocation for the four provinces first increased then decreased. Overall, the annual water allocation increased the most in 2017–2018, by 2.61 billion m^3^, followed by a 1.82 billion m^3^ increase in 2015–2016. The annual water allocation decreased by 0.33 billion m^3^ in 2018–2019 and by 0.78 billion m^3^ in 2019–2020. The water allocated to Beijing decreased the most, by 0.62 billion m^3^ from 2018 to 2020, followed by Hebei province with a decrease of 0.33 billion m^3^ in the same period. Hebei province had the largest water allocation, accounting for 33.96% of the total water allocation in the four provinces. The three other provinces (Henan, Beijing, and Tianjin) accounted for 28.66%, 18.88%, and 18.49%, respectively (Figure 4). The water allocation of Tianjin and Henan provinces increased by 0.1 billion m^3^ and 0.28 billion m^3^ during 2018–2020, respectively. We used the average water price of the cities in their corresponding provinces to estimate the water transfer value in each province from 2014 to 2020. The average first-tier water prices for residents in each province along the SNWD are shown in Figure 5. The water price depended on the cost, the distance that water was transferred, and local policies. We found that the average water price in Beijing was the highest, and the price in Henan was the lowest of the four provinces (Figure 5).

The annual water cost, annual water value of the trunk line project, and four provinces showed variations from 2014 to 2020 (Figure 6). As a whole, the annual water value in each province, except for Beijing, showed an increasing trend. The annual water cost in Henan and Tianjin increased during the study period, and in Beijing and Hebei, it increased in 2019 and then decreased in 2020. The annual water value of the trunk line project in Beijing increased in 2019 and then decreased in 2020, and in Tianjin, it increased throughout the study period. In Hebei, the annual water value of the trunk line project showed a downward trend from 2014 to 2020, and in Henan, it decreased substantially in 2019 and increased slightly in 2020.

From 2014 to 2020, the total water cost, total water value of the trunk line project and four provinces were CNY 4.04, 39.64, and 120.93 billion yuan, respectively (Figure 7). The proportional value split between the three was approximately 1:10:30. Analysis revealed that the total water value increased as the water was transported to the four provinces. The total water value of the trunk line project was about ten times larger than the total water cost, and the total water value of the four provinces was 30 times larger than the total water cost. The profits of China South-to-North Water Diversion Corporation Limited were CNY 35.6 billion yuan, and the profits of water sectors for the four provinces were CNY 81.3 billion yuan, accounting for 69.55% and 30.45% of the total difference, respectively.

### 3.2. Water Transfer of Henan Province

The water allocation, water cost, water value of the trunk line project, and water value of each city in Henan province are shown in Figure 8 and Figure 9. The maximum water allocation of Zhengzhou was 0.54 billion m^3^, which accounted for 18.04% of Henan province allocation. The minimum water allocation of Hua County was 0.05 billion m^3^. The water cost and water value of the trunk line project and cities in Henan province were CNY 0.39, 1.28, and 8.06 billion yuan, respectively (Figure 9). The ratio of the three was approximately 1:3:20, indicating that the SNWD water value increased during transportation. The water value of the trunk line project was about three times larger than the water cost in Henan province, and the water value of Henan province was 20 times larger than the water cost in Henan province. The highest water value of Zhengzhou was CNY 0.07 billion yuan, and the lowest water value of Hua County was CNY 0.007 billion yuan. Xinxiang’s water value in the trunk line project was the highest due to long-distance water transfer and the higher water price at the outlet of the trunk line project than in other cities. Because the water price in Zhengzhou was much higher than that of other cities, and its water allocation was the largest, the water value in Zhengzhou far exceeded that of other cities. The water value of the remaining cities was less than CNY 1 billion yuan.

### 3.3. The Impact of Water Ecosystem Expansion

#### 3.3.1. Spatial-Temporal Variation of Water Ecosystem

The area occupied by the water ecosystem in Xichuan County showed an overall increasing trend from 2000 to 2020 (Figure 10 and Figure 11). During the study period, the total area of the water ecosystem increased by 189.72 km². In 2000–2010, the average area of the water ecosystem was 257.33 km², and that from 2015 to 2020 was 399.13 km², an increase of 141.8 km² compared with 2000–2010. Therefore, the area occupied by the water ecosystem increased rapidly after the SNWD became operational. We used the farmland and urban–rural settlement ecosystem data in 2000, 2010, and 2020 combined with the water ecosystem extraction for Xichuan County to analyze the relative changes in various ecosystems (Table 2). From 2000 to 2010, the expanded Danjiangkou reservoir occupied 63.06 km² of farmland ecosystems and 0.43 km² of urban–rural settlement ecosystems. By 2020, the region had been transformed into a water ecosystem. During the study period, expansion of the water ecosystem accounted for 180.42 km² and 1.25 km² of the farmland ecosystem and urban–rural settlement ecosystem, respectively.

#### 3.3.2. Corresponding Subsequent Subsidy Gap of Ecological Migrants

Here, the subsequent subsidy we mentioned is what the government should pay for the ecological migrant individuals as their living costs in the resettlement region. The minimum standard of living for urban–rural residents, and gaps in subsequent subsidy in each city receiving ecological migrants in Henan province, are shown in Figure 12. By tracking the ecological migrants from Xichuan County, we found they were all moved to other cities in Henan province. The main city that the ecological migrants moved to was Nanyang, which accommodated 96,000 of these new residents, accounting for approximately 59.33% of the ecological migrants. The remaining ecological migrants (40.67%) mainly flowed to Xuchang, Pingdingshan, Zhengzhou, Luohe, Xinxiang, and other cities. We found that the minimum standard of living for urban–rural residents in Zhengzhou was the highest, at CNY 630 yuan/month in urban areas and CNY 430 yuan/month in rural areas (Figure 12). The minimum standard of living for urban–rural residents in Pingdingshan was the lowest, at CNY 520 yuan/month in urban areas and CNY 322 yuan/month in rural areas.

We took the average of the minimum standard of living for urban–rural residents as the compensation standard for other cities in Henan province. Assuming that all ecological migrants were included in the scope of the minimum standard of living for urban–rural residents, calculating the gap in subsequent subsidy for ecological migrants. The subsequent annual subsidy for ecological migrants in Xichuan County was CNY 0.84 billion yuan. The subsequent subsidy gap in Nanyang was the highest, at CNY 0.51 billion yuan, and that in Luohe was the lowest, at CNY 0.02 billion yuan. The results showed that the maximum subsequent subsidy for ecological migrants in Xichuan County accounted for 14.16% of the profits from China South-to-North Water Diversion Corporation Limited, and only accounted for 6.2% of the profits from water sectors in four provinces. Therefore, we defined the difference between the water value of the four provinces and the headwater project as the gross profit, and the subsequent subsidy to ecological migrants accounted for 4.31% of the gross profit. The analysis also revealed that the SNWD and water sectors had sufficient profits to support subsidies for ecological migrants.

## 4. Discussion

This study systematically compared the changes in water cost, water value of the trunk line project, and the four provinces along with the central line project of SNWD from 2014 to 2020. It also analyzed the impacts of ecological migrants of SNWD by estimating the corresponding subsequent subsidy gap that should be provided to the ecological migrants for the first time. Previous studies of the water value of the SNWD showed that most were based on the water transportation cost, over distance, of the SNWD itself [44]. It was reported that the beneficiaries were the organizations or individuals associated with the SNWD (e.g., China South-to-North Water Diversion Corporation Limited and water sectors in each province) [8,13,48]. Although there have been ecological compensations for SNWD, a gap has existed in the corresponding subsequent subsidy and policy for ecological migrants [15,39,49,50,51,52,53]. As one of the most important components in the ecosystem, assessing the impact on humans was indispensable. The resettlement of ecological migrants, and their living conditions after relocation, have been critical factors in evaluating the impacts of SNWD [34,54,55,56]. Our study was novel in considering ecological migrants when assessing the relevant environmental impacts of the water transfer project and estimating the corresponding subsequent subsidy gap for ecological migrants in each city so that an initial quantification of the impact on ecological migrants could be made [53].

Our results also verified that the water value of the SNWD increased during transportation to the four provinces. We found that the direct beneficiaries were always the SNWD project and local water sectors. The profits from China South-to-North Water Diversion Corporation Limited reached CNY 35.6 billion yuan, and the profits from water sectors in four provinces reached CNY 81.3 billion yuan. However, whether the basic compensation for the living costs was able to maintain the standard of living for the ecological migrants of Xichuan County was ignored, and there was no relevant policy to ensure their standard of living in the resettlement region [8,18,34]. Therefore, the subsequent subsidy gap estimated in this study can be used as a reference for future upper subsidy limits for ecological migrants. We took the differences between the total water value of four provinces and the total water cost as the gross profit to evaluate whether the profit can support subsidies for ecological migrants. The maximum corresponding subsequent subsidy gap for ecological migrants in Xichuan County accounted for 4.17% of the annual water value of the four provinces, and that accounted for 4.31% of the gross profit. The results showed that the market income can fully meet the subsequent subsidy gap of ecological migrants, and that the relevant sectors had enough profits to support future subsidization of ecological migrants.

Moreover, as the core water source of SNWD and a county experiencing “state poverty”, Xichuan County has sacrificed development to a certain extent to protect the environment and ensure high-quality water [53]. The water ecosystem in Xichuan County expanded and occupied other ecosystems because of the water supplement when the SNWD became operational. Before the SNWD became operational, Xichuan County officials were very active in addressing environmental pollution and shutdown smelting and chemical enterprises with no compensation. This resulted in the laying-off of 19,000 workers in Xichuan County, and the county’s fiscal revenue decreased by (at its maximum) CNY 895 million yuan from 2014 to 2015 (Figure 13). We carried out this study to estimate the water transfer value and appealed to relevant government sectors to pay attention to the future subsidization of ecological migrants in Xichuan County [20,43,57]. Ecological migrants need to adjust to a new production and living environment after resettlement [33,55]. The resettlement of ecological migrants is closely related to ecological migrants’ development and livelihoods [35,55]. In order to solve the subsequent development problems of ecological migrants, it is effective to provide subsequent subsidies and help them find a job to maintain their livelihoods [35]. The government should track the subsequent flow of ecological migrants and incorporate their future subsidies into the scope of ecological compensation. Our results also indicated that the government should ensure equitable allocation of the profits derived from water use. In addition, the government should further improve the subsequent compensation scheme for ecological migrants. Specifically, the government should gradually improve the subsequent standard and period of ecological migrants. The minimum standard of living for urban–rural residents is a reference for the government when formulating relevant policies, and specific policy formulation should be based on local conditions and the characteristics of ecological migrants [55].

Owing to the absence of specific labor force data, we quantified the impact of ecological migrants by assuming that they were all included in the scope of the minimum standard of living for urban–rural residents. However, the subsequent subsidy gap that we calculated was the upper limit of the subsequent subsidy, which accounted for a small proportion of the water value.

## 5. Conclusions

This study estimated the impacts of ecological migrants and the water value of the SNWD for the first time. The total water cost, total water value of the trunk line project and four provinces were CNY 4.04, 39.64, and 120.93 billion yuan, respectively, and the ratio of the three was approximately 1:10:30. Overall, the water value of the SNWD gradually increased as it was transported to the four provinces. Results proved that the beneficiaries were the China South-to-North Water Diversion Corporation Limited and water sectors in each province. The subsequent subsidy gap we estimated by the minimum standard of living for urban–rural residents only accounted for 4.31% of the gross profit. This indicates that the profits can support the appropriate subsequent subsidy of ecological migrants, and that the needs and concerns of these migrants merit considerably more attention.

## Figures and Tables

**Figure 1 ijerph-18-12295-f001:**
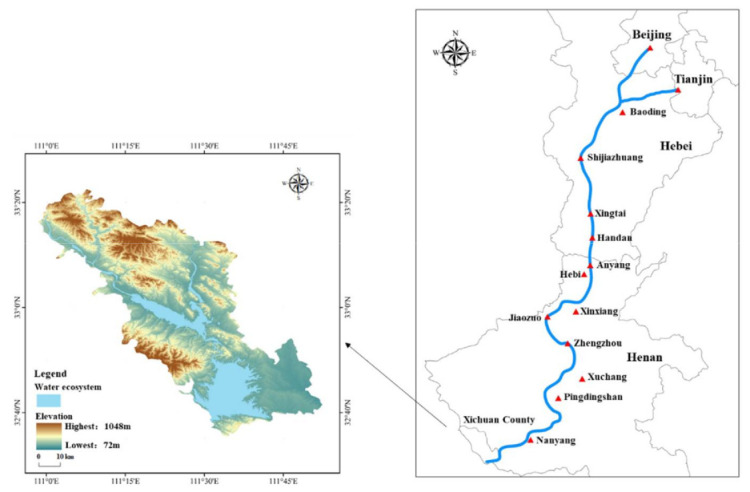
The central line project of SNWD and topographic map of Xichuan County.

**Figure 2 ijerph-18-12295-f002:**
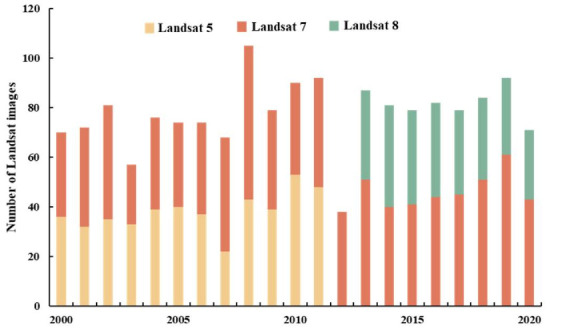
The number of Landsat images.

**Figure 3 ijerph-18-12295-f003:**
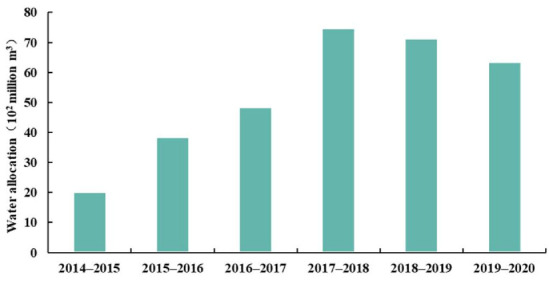
Annual water allocation for four provinces (Beijing, Tianjin, Hebei, and Henan) of SNWD.

**Figure 4 ijerph-18-12295-f004:**
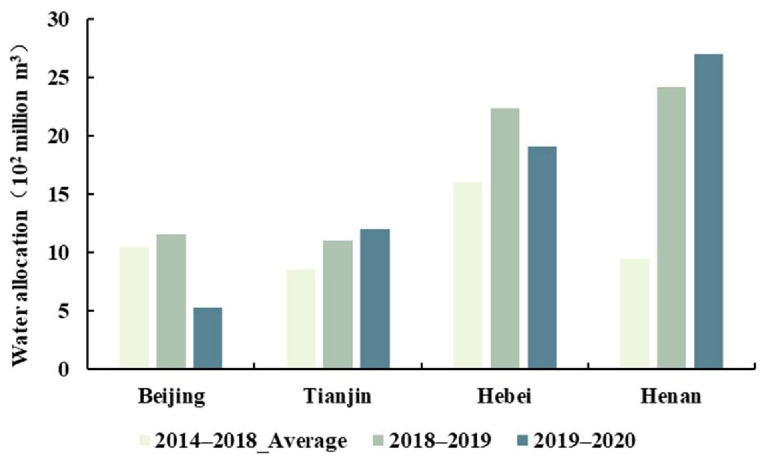
Water allocation for each province of SNWD.

**Figure 5 ijerph-18-12295-f005:**
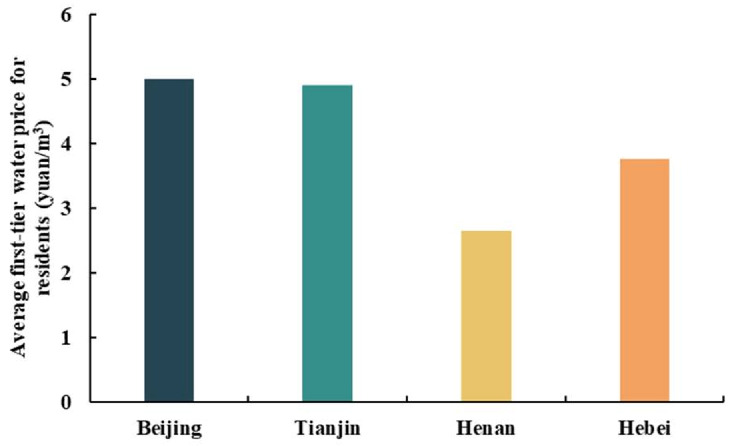
Average first-tier water price for residents of each province of SNWD.

**Figure 6 ijerph-18-12295-f006:**
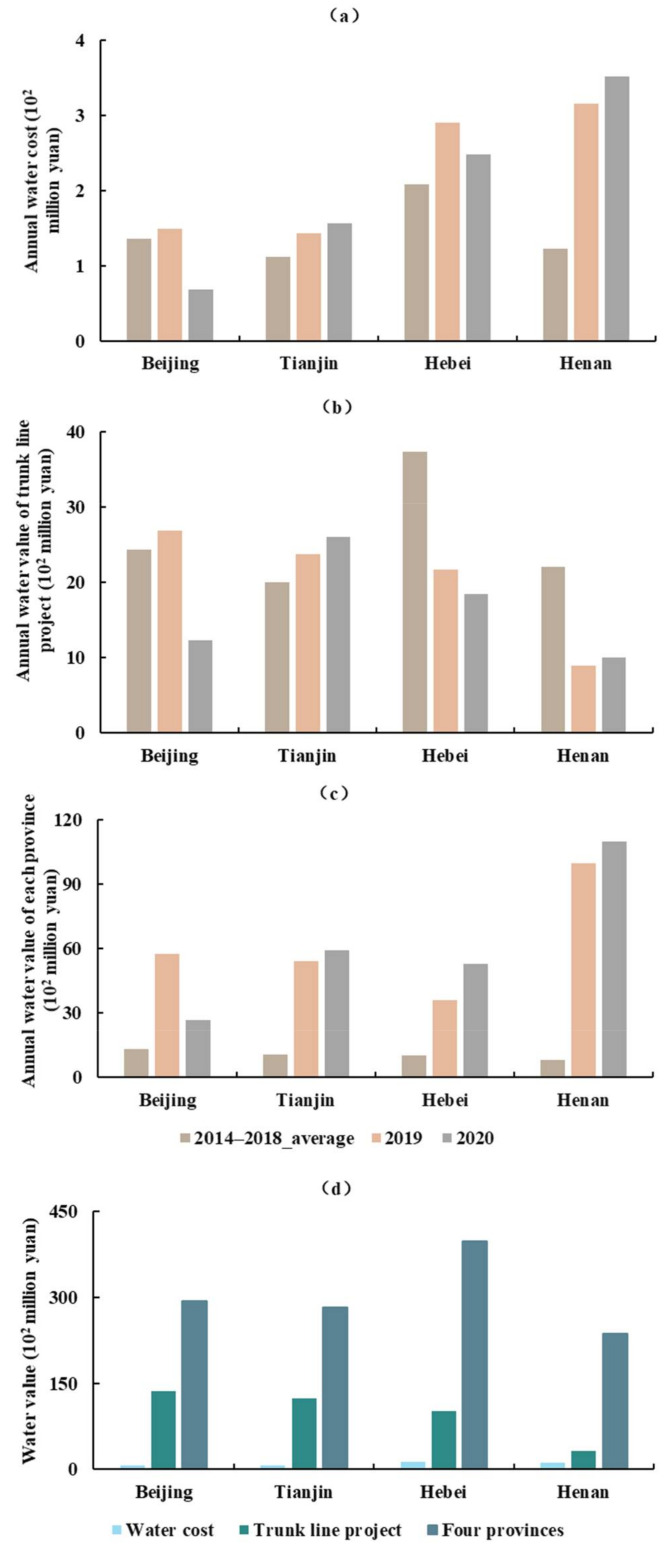
Water value in each province. (**a**) Annual water cost, (**b**) annual water value of the trunk line project, (**c**) annual water value of each province, and (**d**) comparison of water value for the above three.

**Figure 7 ijerph-18-12295-f007:**
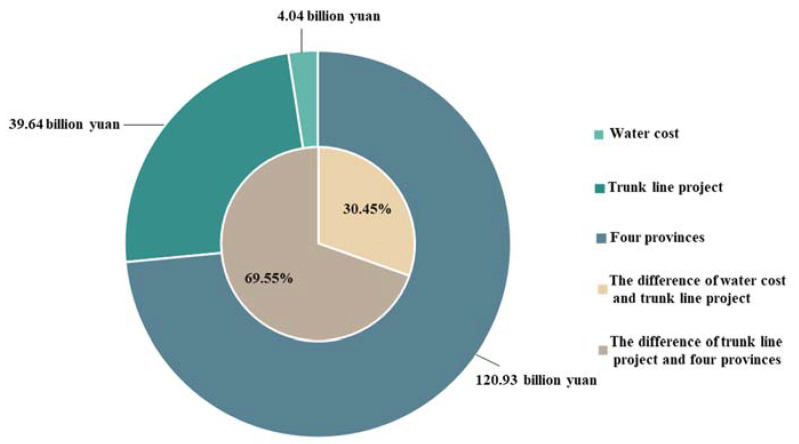
Total water cost, total water value of the trunk line project, and four provinces of the SNWD.

**Figure 8 ijerph-18-12295-f008:**
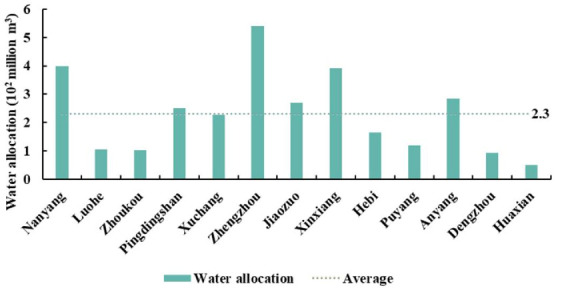
Water allocation of SNWD in Henan province.

**Figure 9 ijerph-18-12295-f009:**
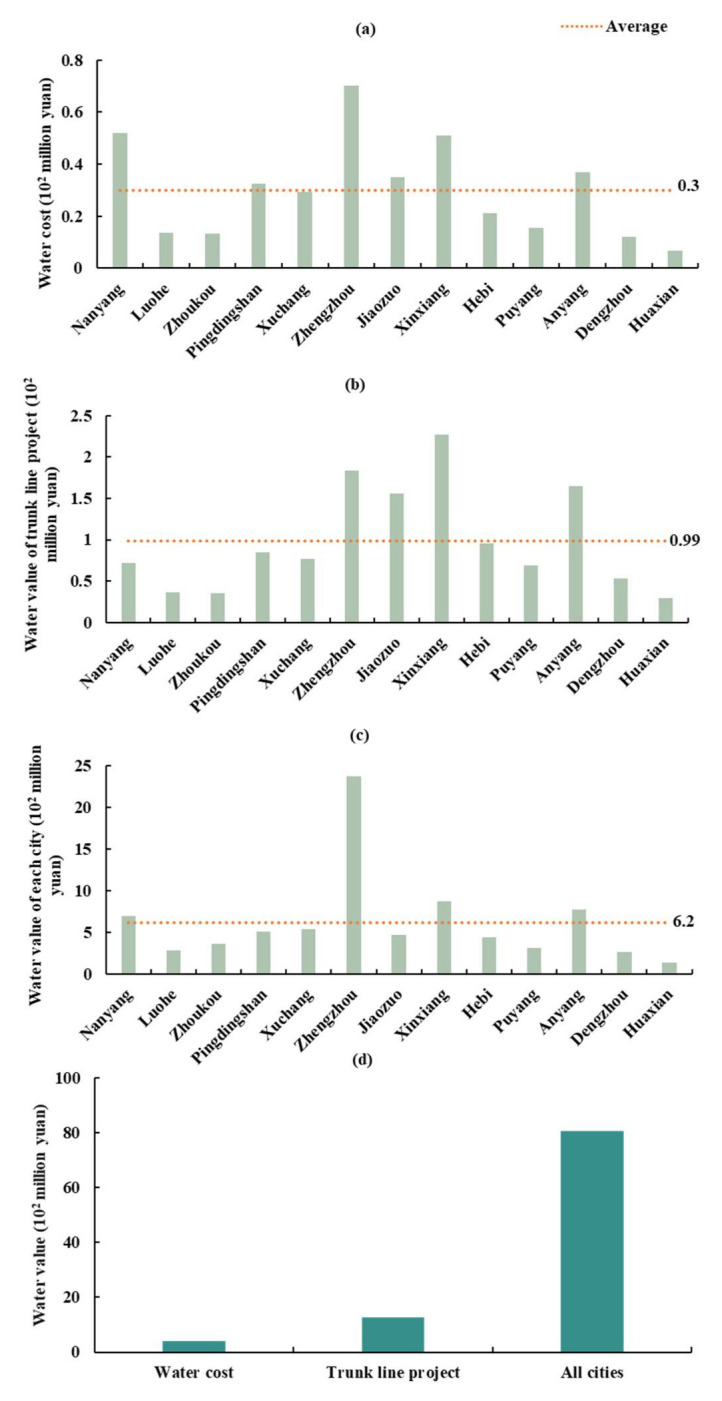
Water value in Henan province. (**a**) Water cost, (**b**) water value of the trunk line project, (**c**) water value of each city, and (**d**) comparison of water value for the above three.

**Figure 10 ijerph-18-12295-f010:**
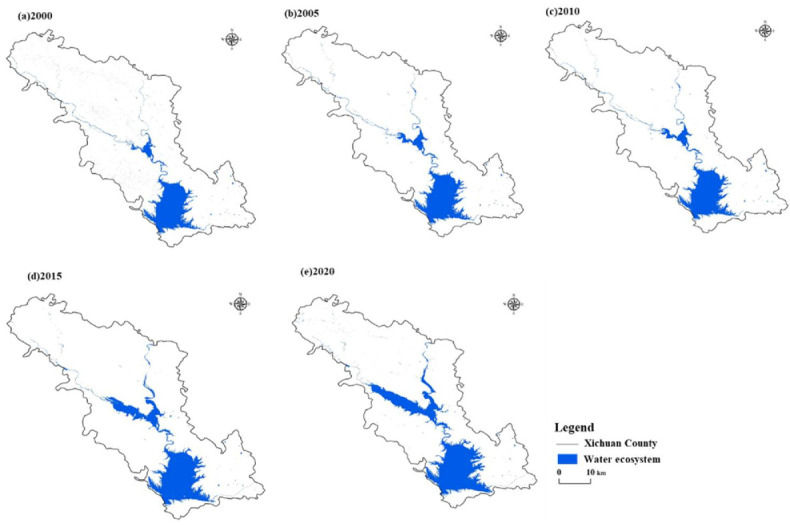
Spatial distribution of annual change for water ecosystem in Xichuan County in (**a**)–(**e**) 2000, 2005, 2010, 2015 and 2020, respectively.

**Figure 11 ijerph-18-12295-f011:**
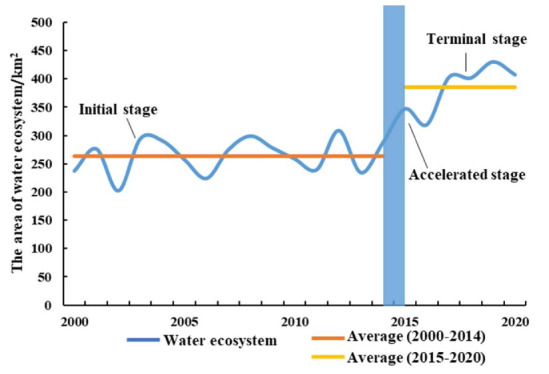
Annual change of water ecosystem area in Xichuan County from 2000 to 2020.

**Figure 12 ijerph-18-12295-f012:**
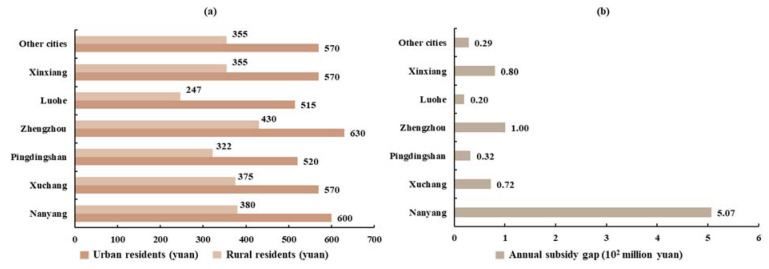
Subsidy of each city. (**a**) Minimum standard of living for urban-rural residents and (**b**) subsequent annual subsidy gap.

**Figure 13 ijerph-18-12295-f013:**
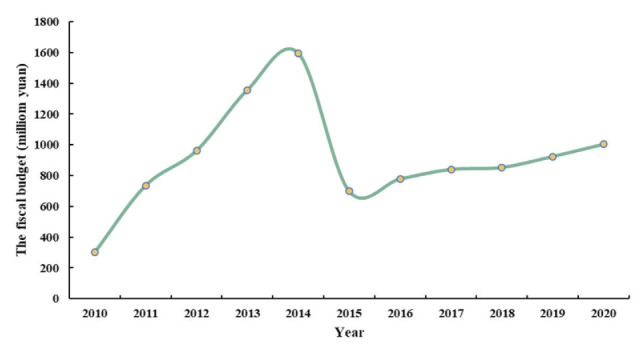
The fiscal budget in Xichuan County from 2010–2020.

**Table 1 ijerph-18-12295-t001:** The outlet water price in the central line project of SNWD.

Section Division		Water Price of Each Section (CNY ¥/m^3^)		
		Comprehensive Water Price	Basic Water Price	Metering Water Price
Headwater project		0.13		
Trunk line project	Nanyang section of Henan province (Wangchenggang—Shili Temple)	0.18	0.09	0.09
	The south section of the Yellow River in Henan province (Xinzhuang—Shangjie)	0.34	0.16	0.18
	The north section of the Yellow River in Henan province (Beileng—Nanliu Temple)	0.58	0.28	0.30
	Hebei province (Yujiadian—Sanchagou) (The south of Langwuzhuang—Deshengkou)	0.97	0.47	0.50
	Tianjin (Wangqingtuo connecting well—Caozhuang pumping station)	2.16	1.04	1.12
	Beijing (Fangshanchengguan—Tuancheng Lake)	2.33	1.12	1.21

**Table 2 ijerph-18-12295-t002:** Distribution of ecosystem around Danjiangkou Reservoir.

	Area (km²)	2000	2010	2020
Type				
Farmland ecosystem		180.42	117.36	0.00
Water ecosystem		191.20	233.90	380.92
Urban-rural settlement ecosystem		1.25	0.82	0.00

## Data Availability

The data presented in this study may be obtained on request from the corresponding author.

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
