# Peer review of "Estimating the Impact of Ecological Migrants on the South-to-North Water Diversion in China"

_ijerph, 2021, doi:10.3390/ijerph182312295_

Round 1

Reviewer 1 Report

This is a very-very interesting paper. I really like the connection of how big infrastructures affects the society in big-scale. 

It also has clear methodology and excellent visualization. 

Congrats.

This paper describes the effect of South-to-North Water Diversion (SNWD) in local societies and economy.

I find this paper very interesting as it connects a large-scale project (which transforms the distribution of Chinas’ natural resources) to the effects of society and ecological migrants.

Introduction is well documented with many references and describes efficient the paper.

Study area is clear presented with data and visualization.

Data sources are presented and shows that authors had done a very serious and systematic research.

Methods are also clearly presented but I have not use them in my research, to validate them (i.e. I have not study something with these methods). However, they seem appropriated.

The results are clear with a very good visualization.

I was very impressed by Figure 11.

Based on the data, and the representation in diagrams, the conclusions are logical. 

Reviewer 2 Report

The study has focused on water security and its impact on ecological migrants. This is an important subject in environmental policy and water use strategy. The manuscript is well written but there is room for more improvement especially in the areas of the review of literature, streamlined objectives and the discussion of results. Specific comments are the following:

INTRODUCTION

- Page 1, lines 38-39. As a result, the groundwater level in Beijing rose from 25.7 m in 2014 to 38 22.7 m in 2019, since the SNWD became operational [9]. Dos the authors intend to say declined from 25.7m to 22.7m?

- Page 2, lines 61-62. Provide reference to support the claim made.

- This study has focused on the impact of development and construction on ecological migrants.  Provide an extensive literature review on the impact of development projects on ecological migrants and how adverse impact can be addressed.

- The objective is not clear. Please, provide a streamlined objectives and potential contribution of the findings from the study.

DISCUSSION

- Page 11, lines 307-308. Provide references of the studies being referred to here.

- Provide a detailed discussion of the results by telling the reader whether the results are in line with the findings in the literature on the subject. If the results are not in line with findings in the literature, provide the potential reasons for the difference.

- Discuss the implications of the results from this study on relevant policies and strategies.

- Discuss how the issue associated with ecological migrant can be addressed.

CONCLUSION

- The current version of the conclusion section appears to lack the most important conclusions that can be drawn from the results. Provide the most important conclusions that can be drawn from the results from this study.

Reviewer 3 Report

see uploaded file

Round 2

Reviewer 2 Report

Some the concerns raised in the earlier version of the manuscript has not been addressed. Address the following concerns: Discussion section 1. Discuss the implications of the results from this study on relevant policies and strategies. 2. Discuss how the issue associated with ecological migrant can be addressed. CONCLUSION The current version of the conclusion section appears to lack the most important conclusions that can be drawn from the results. Provide the most important conclusions that can be drawn from the results from this study. Aim of the study Our study aimed to provide scientific data for the relevant sectors on the water 88 transfer value of the SNWD and the subsequent subsidy policy regarding ecological 89 migrants. This is not very clear. The aim/objectives should be related to the title of the manuscript. Rewrite the aim(s)/objective(s) to capture the subject of manuscript. It should reflect the title. Note that when submitting response report, provide page numbers and line number where changes have been made. Areas where changes have made should be in track changes.

Reviewer 3 Report

This reviewer thanks the authors for their clarifications, all of which helped to clarify a number of issues. It would be appropriate to incorporate the minor adjustments to the paper, as described in the attached file. Otherwise, in my opinion, paper is very interesting and warrants publication
